

# Study of NO₂ and HCHO vertical profile measurement based on Fast Synchronous MAX-DOAS

Jiangman Xu[1,2], Ang Li[1,*], Zhaokun Hu[1], Hairong Zhang[1,2], Min Qin[1]

[1] Key Laboratory of Environmental Optics and Technology, Anhui Institute of Optics and Fine Mechanics,

Hefei Institutes of Physical Science, Chinese Academy of Sciences, Hefei 230031, China

[2] University of Science and Technology of China, Hefei 230026, China

*Correspondence to*: Ang Li (angli@aiofm.ac.cn)

**Abstract**: This study investigates a multi-elevation Fast Synchronous Multi-Axis Differential Optical

Absorption Spectroscopy (FS MAX-DOAS) observation system that can rapidly acquire trace gas

profiles. It modifies the conventional MAX-DOAS method by sequentially scanning at elevation angles

using motors. The new system incorporates a two-dimensional area array Charge Coupled Device (CCD)

grating spectrometer, small field-of-view telescopes (<1°), a high-speed shutter switching module, and a

multi-mode multi-core fiber to enable multi-channel spectroscopy and significantly enhance the time

resolution of the collected spectra (one elevation cycle within two minutes). When selecting the

spectrometer grating, the impact of spectral resolution on the detection of nitrogen dioxide (NO₂) and

formaldehyde (HCHO) by FS MAX-DOAS was simulated and analyzed. The optimal resolution range

was determined to be 0.3-0.6nm. The selection of the number of binning rows in the acquisition settings

considers the signal-to-noise ratio of the pixels in each row to enhance the quality of the spectral data.

Two-step acquisition is used for low-elevation angles within one cycle to overcome the influence of

variations in light intensity. A comparative test was conducted on outfield NO₂ and HCHO measurements

using differential optical absorption spectroscopy. Compared with the differential slant column

densities(dSCDs) at each elevation angle measured by the MAX-DOAS system, the Pearson correlation

coefficient of NO₂ reached 0.9, while for HCHO it ranged mostly between 0.76 and 0.85. The results of

the slant column concentration inversion indicate that the root mean square (RMS) of the FS MAX-

DOAS spectrum inversion can consistently be lower than that of MAX-DOAS over an extended period.

The profile results show that the diurnal variation trend of the two systems was consistent, and because

of the enhanced time resolution, the gas profile obtained by the former system can provide more detailed

information. Compared with the near-ground NO₂ concentration measured by the long-path DOAS

system, the daily variation trend shows a characteristic of being high in the morning and starting to

decrease at noon, and the correlation coefficient between FS MAX-DOAS and LP -DOAS is higher (R



= 0.880). The FS MAX-DOAS system can quickly and simultaneously obtain the vertical distribution profiles of NO₂ and HCHO with high accuracy, providing a basis for mobile MAX-DOAS to achieve gas profile inversion.

### 1.Introduction

Nitrogen dioxide (NO₂) is a toxic gas with a pungent odor and is a key pollutant that affects ambient air quality and human health(Jion et al., 2023). NO₂ can be generated from ozone, and in recent years, its surface concentration in urban areas has increased, leading to an elevated risk of respiratory diseases and other serious health conditions(Kuerban et al., 2020). Emissions from industrial activities have exacerbated NO₂ pollution. The main sources of NO₂ are fossil fuel combustion, vehicle exhaust, and

power generation. Formaldehyde (HCHO) is a carbonyl compound that triggers photochemical reactions in air pollutants. The lifespan of HCHO is very short in the troposphere, and its photolysis produces hydroxyl radicals (OH), which drive the photooxidation process and ultimately lead to the formation of ozone(Lui, 2017). These pollution gases have caused numerous adverse effects in the ambient atmosphere.

At present, detection methods for NO₂ and HCHO mainly include electrochemical sensors, Fourier Transform Infrared Spectroscopy (FTIR), Laser-Induced Fluorescence Spectroscopy (LIF), Differential Absorption Lidar Technology (DIAL), and Differential Optical Absorption Spectroscopy (DOAS). Traditional chemical methods can be traced back to the 1950s; however, these methods are currently not widely used due to certain deficiencies in data quality. The University of Liège in Belgium(Franco et al.,

2015) used ground-based FTIR and MAX-DOAS to obtain the HCHO profile at a station in the Swiss Alps. They also employed a chemical transport model to simulate the total column volume and compared it with the data from the two instruments. The University of L'Aquila(Di Carlo et al., 2013) developed a thermal dissociation laser-induced fluorescence instrument to measure NO₂ and compared it with a chemiluminescence system to evaluate the instrument's performance. The DOAS technology is a non-

contact measurement method with a broad measurement range and an extensive monitoring range, capable of simultaneously measuring multiple gases. The Korea Advanced Institute of Science and Technology(Lee et al., 2005) verified the feasibility of the LP-DOAS system for the simultaneous measurement of NO₂, SO₂, and HCHO in Asian urban areas, with a detection limit at the ppb level. As a passive DOAS technology, MAX-DOAS uses sunlight as the light source, which makes it easy to operate

and detect the vertical profiles of multiple gases simultaneously. The Heidelberg University in



Germany(Hönninger and Platt, 2002) took the lead in using MAX-DOAS for the first time in Canada's ALERT2000 polar sunrise experiment. In this experiment, the observation mode of each off-axis angle was introduced based on the zenith angle to obtain the vertical profile of BrO and it was found that dSCDs decreased with the increase in elevation angle, which can be traced back to an earlier study by

Sanders et al(Sanders et al., 1993), who used off-axis geometry to observe stratospheric OClO in Antarctica. Roland J. Leigh(Leigh et al., 2006)from the University of Leicester in the UK developed a device for synchronous observation at off-axis angles to monitor rapidly changing urban $NO_2$ concentration, which improved the time resolution and proved the potential of synchronous observation method to gather information on rapidly changing $NO_2$ concentrations and spatial distribution in the

environment. However, he did not investigate the vertical profile distribution of $NO_2$ any further. In recent years, research on MAX-DOAS gas profile inversion has matured. Wang Y et al(Wang Yang et al., 2013) studied a method based on the optimal estimation method (OE, first proposed by Frieß et al in 2006) to obtain the vertical profile and vertical column concentration of tropospheric $NO_2$. Wagner(Wagner et al., 2011) used the look-up table method (first proposed by Li Xin from Peking University in 2010 ) to invert

the profile distribution of $NO_2$ and HCHO in the troposphere in Milan during the summer of 2003. However, the configuration of the MAX -DOAS system, which depends on stepper motors to adjust the telescope angle, has hardly changed(Heckel et al., 2005; Tian et al., 2019; Wang et al., 2017). The efficiency of spectral collection in MAX-DOAS needs improvement, and enhancing the temporal resolution of MAX-DOAS is crucial for studying the gas profile distribution using mobile MAX-DOAS.

Table 1 presents the time resolutions of various MAX-DOAS equipment worldwide in recent years, which are generally low.

**Table 1 The time resolution of some MAX-DOAS equipment in the world**

| Location | Instrument parameter | Targets | Temporal resolution | Reference |
|---|---|---|---|---|
| India | Angels:1°,2°,3°,5°,10°,20°,40°,90°, FWHM=0.7nm | CHOCHO | 9min | Mriganka Sekhar Biswas et al.(2023)(Biswas et al., 2023) |
| Vienna | Angels:1°,2°,3°,4°,5°,10°,15°,30°, 90°, FWHM=0.7nm | $NO_2$,HCHO, CHOCHO | 6min | Stefen F.Schreier et al.(2020)(Schreier et al., |



| | | | | |
|---|---|---|---|---|
| | | | | 2020) |
| Shanghai | Angels:5°,10°,15°,20°,90°, | $NO_2$ | 5-15min | K.L. Chan(2015)(Chan et |
| | FWHM=0.6nm | | | al., 2015) |
| Hong Kong | Angels:3°,4°,5°,6°,8°,10°,15°, | $NO_2,O_4$ | 10min | M. Wiegner(2018)(Chan |
| | 30°,45°,90°, FWHM=0.7nm | | | et al., 2018) |

In this study, we introduce a fast synchronous MAX-DOAS system capable of swiftly acquiring trace gas profiles. Combined with the concept of multi-channel spectroscopy, simultaneous observations at multiple elevation angles are conducted, significantly enhancing the temporal resolution of spectrum acquisition. The selection of the number of binning rows in CCD takes into account the signal-to-noise ratio of each row of pixels to improve spectral quality. A two-step acquisition strategy is adopted in the acquisition program to mitigate the differences in light intensity between high and low-elevation angles. The performance of the established FS MAX-DOAS system was verified through comparative experiments with ground-based MAX-DOAS and LP-DOAS for $NO_2$ and HCHO detection. The RMS of the new system's spectral inversion was consistently lower than that of MAX-DOAS for an extended period. The gas profile of the FS MAX-DOAS can also provide more information and the FS MAX-DOAS system can quickly and accurately obtain the vertical distribution profiles of $NO_2$ and HCHO simultaneously.

## 2. Measurement principle

### 2.1. Dispersive spectrometer with Binning technology

In this study, an IsoPlane series spectrometer (Princeton Instruments) was used. The core spectroscopic element is a grooved grating with high diffraction efficiency(Zhu Jiacheng et al., 2017). It features a patented astigmatism-free Schmidt-Czerny-Turner design spectroscopic structure. Compared with the typical Czerny-Turner optical path structure(You et al., 2004), the astigmatism on the focal plane is eliminated, and the optical aberration is significantly reduced. The spectral signal receiving part consists of a two-dimensional area array CCD. The CCD pixel binning technology(Westra et al., 2009) can greatly reduce the spectrum collection time. Binning occurs in the charge domain, making it essentially a noise-free operation. This leads to lower readout noise and an increased signal in the register due to the sum of multiple pixel signals. Consequently, this leads to a higher spectral signal-to-noise ratio and faster signal output rate. However, the disadvantage is that a certain spectral resolution is sacrificed.

### 2.2. MAX-DOAS retrieval



MAX-DOAS technology combines multiple observation elevation angles to measure the atmospheric
spectrum. It is different from active DOAS technology, which uses scattered solar light as the light source.
This technology considers multiple scattering radiation transfer models and aerosol conditions and is
often used to study tropospheric trace gases such as $NO_2$, BrO, $SO_2$, HCHO, and $H_2O$. The principle basis
of this technology for quantitative gas analysis is Lambert-Beer's law(Platt and Stutz, 2008).

$$I(\lambda) = I_0(\lambda) \bullet \exp[\sum_{j=1}^{n} -\sigma^j(\lambda) \bullet c_j \bullet L] \tag{1}$$

In the formula $I(\lambda)$, $I_0(\lambda)$, $\sigma^j(\lambda)$, and $C_j$ respectively represent the measured spectrum after
atmospheric absorption and extinction, the zenith spectrum without extinction, the gas absorption cross
section and column density, L represents the optical path of light as it travels through the absorbing gas.
According to the initial step of spectral analysis in the MAX-DOAS method inversion process, this study
used QDOAS software to calculate the differential slant column concentration of trace gases ([http://uv-vis.aeronomie.be/software/QDOAS/](http://uv-vis.aeronomie.be/software/QDOAS/) ). This process uses the least-squares fitting algorithm to solve the
differential slant column concentration of gases. It employs low-order polynomials to represent Rayleigh
scattering, Mie scattering and other broadband attenuation, with the primary focus being on analyzing
the contribution of narrow-band spectral structure characteristic absorption to optical thickness. Since
dSCDs depend on the geometry of the observation, sun position, cloud interference, aerosol load, surface
reflectivity, etc. It cannot intuitively represent the spatial information of the gas and usually needs to be
converted into vertical column density (VCD)(Rozanov and Rozanov, 2010).

$$VCD = \frac{DSCD}{DAMF} = \frac{dSCD_{\alpha \neq 90^{\circ}} - dSCD_{\alpha = 90^{\circ}}}{AMF_{\alpha \neq 90^{\circ}} - AMF_{\alpha = 90^{\circ}}} \tag{2}$$

where AMF is the atmospheric mass factor, usually determined through the atmospheric radiative transfer
model (RTM). This study uses the PriAM algorithm(Wang et al., 2018; Wang Yang et al., 2013) jointly
developed by the Anhui Institute of Optics and Fine Mechanics and the Max Planck Institute of Chemistry,
to invert the column concentration and vertical profile of the gases. This algorithm is based on the optimal
estimation algorithm and combined with the SCIATRAN model. It considers the sensitivity of the gas
slant column concentration at different viewing angles to the atmosphere at various altitudes, enabling
the inversion of the vertical profile of the aerosol extinction coefficient and gas volume mixing
ratio(VMR). Since aerosols have a significant impact on the transmission path of light in the atmosphere,



the first step of the algorithm involves calculating the weight function $K_{O4}$ that corresponds to the

measurement state through RTM. This function is then used as input for the optimization method, along

with the dSCDs from multiple altitude angles and gas priori profiles, to calculate the vertical profile of

aerosol extinction coefficient and aerosol optical depth(AOD). The second step is to input the aerosol

profile, aerosol single scattering albedo and asymmetry factor into the RTM. The gas concentration

vertical distribution inversion algorithm is used to obtain the gas tropospheric (0-4km ) VMR vertical

distribution profile and VCD(Ren et al., 2021; Wang et al., 2017).

### 3.Parameter analysis

In the study of MAX-DOAS detection of trace gases, the spectral resolution is directly related to the

accuracy of the DOAS inversion. The FS MAX-DOAS system we constructed aims to simultaneously

obtain high-temporal resolution profiles of $NO_2$ and HCHO. This section analyzes the optimal resolution

for detecting the two gases simultaneously. The characteristic absorption bands of the gases under study

gases are located in different spectral bands. Combined with the spectal band of the FS MAX-DOAS,

the optimal UV inversion bands for the two gases were determined based on the recommended

configuration of the international CINDI campaign, HCHO(336.5-359nm)(Pinardi et al., 2013) and

$NO_2$(338-370nm)(Roscoe et al., 2010). The high-resolution standard absorption cross sections for trace

gas analysis were obtained from the MPI-Mainz UV/VIS database. The QDOAS software can

automatically identify the resolution of the entire detection spectrum band and convolved it with the

standard absorption cross-section σ(λ) to achieve the same resolution level as the spectrum being

analyzed. The differential absorption cross section σ'(λ) can be expressed as

$$\sigma'(\lambda) = (G * \sigma)(\lambda) = \int \sigma(\lambda') \cdot G(\lambda - \lambda') d\lambda' \qquad (3)$$

In the formula, G(x) is a Gaussian instrument function.

### 3.1 Effects of resolution on absorption characteristics

σ'(λ) is obtained by convolving the high-resolution gas standard absorption cross-section with the

instrument function of different spectral resolutions, as shown in Fig 1. As the σ'(λ) of $NO_2$ and HCHO

continue to increase with $\Gamma_0$, the shapes of their characteristic absorption structures change. The main

manifestations are as follows: there is a shift in the position where the strongest absorption structure

appears; the characteristic absorption of the gas molecules becomes flat at lower resolutions, and the fine

feature absorption at high resolutions gradually disappears; simultaneously, a decrease in resolution also





reduces the differential absorption cross-section value, and the overall height of the characteristic

absorption bands decreases.

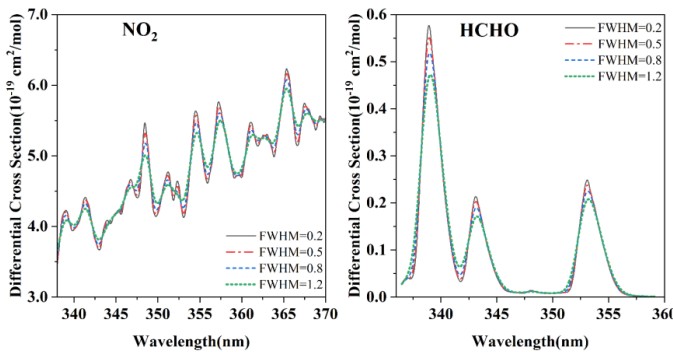

**Figure 1 Differential absorption cross sections of NO₂ and HCHO at different spectral resolutions**

**3.2 Effects of resolution on differential absorption cross-section**

DOAS technology mainly measures the differential optical thickness (D') of the gas. The D' of gas as a

function of resolution $\Gamma_0$ is proportional to its differential absorption cross-section $\sigma'(\lambda)$. As $\Gamma_0$ increases,

the value of $\sigma'(\lambda)$ continuously decreases, and due to the differences in the characteristic absorption of

different gases across different bands, their trends change with $\Gamma_0$ differ. Select two wavelengths with

strong absorption for the two gases being measured, and calculate the relative changes in the differential

absorption cross-sections at different resolutions. Figure 2 shows the changing trend of the maximum

$\sigma'(\lambda)$ of the gas with $\Gamma_0$ at the selected wavelengths.

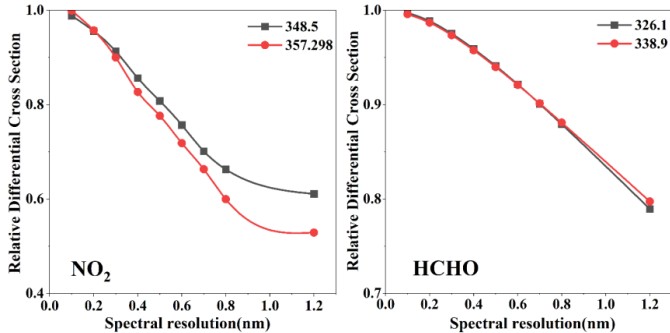

**Figure 2 The changing trend of differential absorption cross-section of NO₂ and HCHO with resolution**

It can be seen from the figure that the slope changes of the two curves of HCHO are basically the same,

indicating that the influence of $\Gamma_0$ on the differential absorption cross-section value $\sigma'(\lambda)$ of HCHO is



consistent at different wavelengths. The slope change of the $NO_2$ curve is significantly larger. The influences of $\Gamma_0$ on $\sigma'(\lambda)$ vary significantly at different wavelengths. For $NO_2$, when $\Gamma_0$ is greater than 1 nm, the decreasing trend of $\sigma'(\lambda)$ slows down. Therefore, even the same gas is affected differently by the resolution at different absorption wavelengths. Although the change in $\sigma'(\lambda)$ with $\Gamma_0$ is more complicated,

in general, the differential absorption cross-section value decreases as the resolution value increases. Ignoring the influence of other factors, the minimum detection limit of the gas concentration is:

$$c_{min} = \frac{D_0}{\sigma'(\lambda, \Gamma_0) \cdot L} \tag{4}$$

where, $c_{min}$ is the minimum detectable concentration of the gas, $D_0$ is the minimum detectable optical thickness, and $\sigma'(\lambda, \Gamma_0)$ is the differential absorption cross-section of the gas at a certain wavelength

and resolution. $\sigma'(\lambda)$ decreases as $\Gamma_0$ increases. Combined with Formula 4, the detection limit of gas decreases as $\Gamma_0$ increases. Therefore, in order to reduce the detection limit, the system should ensure a smaller $\Gamma_0$, that is, a higher spectral resolution.

**3.3 Effects of resolution on signal-to-noise ratio**

Theoretically, as long as $\sigma'(\Gamma_0)$ is any function of $\Gamma_0$ (it decreases as $\Gamma_0$ increases), $\sigma'(\Gamma_0)$ can be

approximated as a linear function of $\Gamma_0$:

$$\sigma'(\Gamma_0) = f(\Gamma_0) \approx \sigma \cdot (1 - b \cdot \Gamma_0) \tag{5}$$

where $b$ is a constant. The relationship between signal-to-noise ratio $D'/N$ and $\Gamma_0$ can be obtained:

$$D'/N = \sigma \cdot (1 - b \cdot \Gamma_0) \cdot \Gamma_0 \propto \Gamma_0 - b \cdot \Gamma_0^2 \tag{6}$$

Among them, $N = D_0$, the SNR has a quadratic function relationship with $\Gamma_0$. In order to obtain the best

SNR, the derivative of the quadratic function can be set to 0, and the optimal resolution $\Gamma_{0opt}$ is:

$$\Gamma_{0opt} = \frac{1}{2 \cdot b} \tag{7}$$

The values of $b$ of the measured gas at different wavelengths were obtained through linear fitting. The fitting process is explained using 348.6nm of $NO_2$ as an example in Fig 3. The calculation of the optimal resolution for the other gases is similar. The $b$ values of $NO_2$ and HCHO in different wavebands and

their corresponding $\Gamma_{0opt}$ values obtained by fitting are listed in Table 2. If two gases are detected simultaneously, the system should ensure that the spectral resolution is optimally between 0.2-



0.6nm .

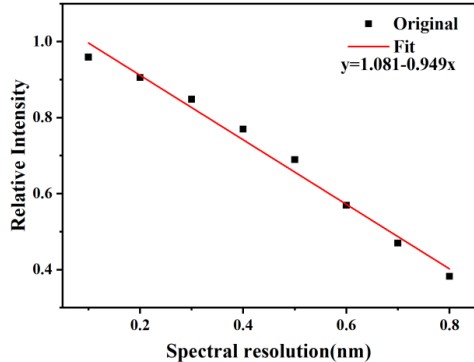

**Figure 3 Linear fitting of NO₂** $\sigma'(\Gamma_0)$ **and Γ₀ at 348.6nm**

**Table 2 Linear relationship between the differential absorption cross-section and spectral resolution**

| Compounds | Wavelength(nm) | Linear regression | b/nm⁻¹ | Γ₀ opt/nm |
|-----------|----------------|-------------------|--------|-----------|
| NO₂ | 340.8 | Y=0.896-0.802x | 0.895 | 0.56 |
| | 368.5 | Y=0.804-1.856x | 2.308 | 0.22 |
| HCHO | 336.5 | Y=0.698-1.256x | 1.799 | 0.28 |
| | 358.1 | Y=1.124-1.348x | 1.199 | 0.42 |

**4. Experimental system, parameters and acquisition program**

**4.1 Experimental system and parameter configuration**

**4.1.1 Experimental device**

The Fast Synchronous MAX-DOAS(FS MAX-DOAS) we built is illustrated in Fig 4. The spectrometer used was the IsoPlane series from Princeton Instruments. We acquired a diffraction grating that fulfills the specifications for wavelength range and resolution. We used Princeton Instrument's fully integrated low-noise camera PIXIS series, which features thermoelectric cooling to -75 ℃ to effectively minimize dark current noise. It has a built-in area array CCD with pixel specifications of 512 × 2048 pixels. The

ultraviolet anti-attenuation fiber used was also specially designed, as shown in Fig 5. The overall structure features a multi-mode multi-core design(Takenaga et al., 2011). The front end of the spectrometer's slit was connected to form a bundle, while the other end connected to the telescope was divided into twelve bundles. The twelve split beams are divided into three groups: B, C and D; one group



includes four beams, as shown in the fiber end face in Fig 6. Each split bundle of class C and D integrates

two cores hat correspond to a viewing angle. In other words, each viewing angle can illuminate two

binning areas in the CCD; in order to ensure that each viewing angle spectrum has its corresponding 90°

spectrum, four cores were integrated into each class B beam, corresponding to two viewing angles. The

A-end connected to the spectrometer was combined and divided into four small areas to maximize the

use of the CCD imaging surface. The eight inner cores in each area were arranged in a crossed Y-shaped

structure. The fiber design meets the requirement that the spectra of low angles and their corresponding

90° responses originate from the same CCD region. This design helps suppress the differences in the

spectral structure caused by variations in CCD performance. To achieve multi-angle simultaneous

collection, telescopes that receive scattered light must be redesigned. We designed a compact and

lightweight achromatic triplet lens optimized for the UV band (290-400nm)(Tang et al., 2020). In order

to avoid crosstalk between signals at various angles, the field of view angle needs to be within 1°, and

the actual field of view angle range of the designed lens is 0.52-0.72°, which satisfies the experimental

requirements. We have also developed a multi-channel electromagnet mechanical shutter control system

to switch between the low-angle and 90°angle telescopes. The computer sent instructions to the

microcontroller following the established serial communication protocol. A shutter corresponds to a byte

address and is controlled independently. The control byte has only two states: ON and OFF. In this way,

the shutter was controlled to open at the required angle to receive the optical signal. This allowed for

obtaining spectra from both low angles and 90° angles without interference between them.

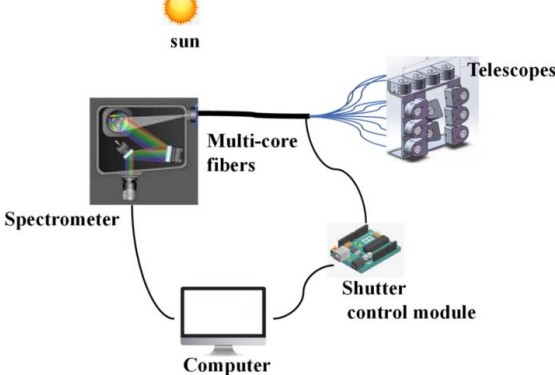

**Figure 4 Schematic diagram of FS MAX-DOAS system structure**



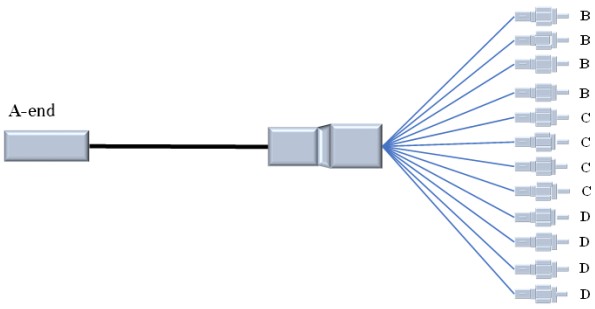


**Figure 5 Appearance diagram of optical fiber**

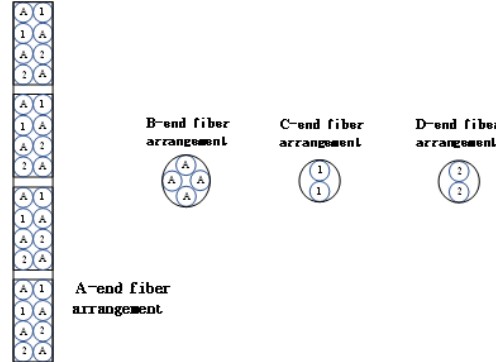

**Figure 6 Optical fiber arrangement at A, B, C, and D ends**

**4.1.2 Optical parameters of each channel**

We use the idea of multi-channel spectroscopy to partition the entire area array CCD into multiple

sections, creating independent spectral detection systems with a spectrometer and telescopes. Therefore,

it is necessary to evaluate the optical parameters of each channel, and according to the analysis in Section

3.3, the optimal resolution ranges for $NO_2$ and HCHO inversions are different. When selecting the grating

specification and slit width for the experiments, a comprehensive consideration must be taken to select a

configuration that can simultaneously satisfy the inversion of both gases. The spectral resolution of a

fiber spectrometer system is determined by the fiber core diameter, the number of grating lines, and the

slit width. In theory, when the fiber core diameter is determined, the greater the number of grating lines

and the narrower the slit, the higher the resolution(Li et al., 2020). The system used an 1800g/m grating

and a slit width of 300um. The optical parameters of each channel are tested, as listed in Table 3.

Combined with the analysis of the optimal resolution for detecting two gases in Section 3, the resolution

of each channel of this system covers a range of 0.3-0.6nm, meeting the requirements for the



simultaneous detection of the two gases.

**Table 3 Optical parameters of each optical fiber core**

| Fiber number | FOV(°) | range(nm) | Resolution(nm) |
|:---:|:---:|:---:|:---:|
| C1-1(5°) | 0.6112 | 307.1807-371.5234 | 0.39-0.58 |
| C1-2(5°) | 0.6112 | 306.5083-370.8228 | 0.31-0.61 |
| C2-1(10°) | 0.6112 | 306.5376-370.8599 | 0.27-0.53 |
| C2-2(10°) | 0.6112 | 306.6827-370.8126 | 0.30-0.61 |
| C3-1(15°) | 0.6112 | 306.6208-370.8902 | 0.32-0.59 |
| C3-2(15°) | 0.6112 | 307.1971-371.5428 | 0.43-0.56 |
| C4-1(1°) | 0.5278 | 307.1646-371.4931 | 0.40-0.54 |
| C4-2(1°) | 0.5278 | 307.1782-371.4896 | 0.43-0.58 |
| D1-1(30°) | 0.6388 | 307.1970-371.5342 | 0.35-0.52 |
| D1-2(30°) | 0.6388 | 306.6421-370.9483 | 0.34-0.60 |
| D2-1(8°) | 0.6112 | 306.5851-370.8193 | 0.42-0.60 |
| D2-2(8°) | 0.6112 | 307.2105-371.5200 | 0.40-0.58 |
| D3-1(2°) | 0.5833 | 307.2620-371.6036 | 0.37-0.58 |
| D3-2(2°) | 0.5833 | 306.5376-370.8126 | 0.45-0.59 |
| D4-1(3°) | 0.5833 | 307.2620-371.6036 | 0.35-0.53 |

**4.2 Collection control method**

**4.2.1 Number selection of binning rows**

An excessively large binning area can easily lead to the inclusion of spectra from other neighboring angles, thereby impacting the quality of the spectrum within that area. It was necessary to select the number of qualified rows according to the SNR for binning .

For the receiving module CCD camera in the FS MAX -DOAS system , the SNR is an indicator of the imaging quality(Cota et al., 2009). During the CCD imaging process, in addition to the real signal, a series of uncertain noises are also introduced(Uncertainty of the optical signal itself, thermal motion of electrons, electronic noise, etc.). The first part is the shot noise generated by the uncertainty in the optical signal, the second part is the dark current due to the thermal motion of the electrons; and the third part is the readout noise caused by the signal interference from the on-chip amplifier(Wang et al., 2013). In the



experiment, the CCD was cooled to -70 ℃ to reduce the dark current noise.

According to the theoretical SNR analysis, quantitative analysis is still necessary to enhance the spectral

quality and improve the SNR of CCD cameras. In order to facilitate signal collection, experimental

analysis, calculations, and specific tools were used. The signal value output from the fiber spectrometer

system is a value mixed with noise. Therefore, an approximate calculation method was used to determine

the corrected average value of the output signal, which represents the signal effective value $\overline{U}_f$ and the

average value of the signal jitter corresponding to the noise value $\overline{C}$, then the SNR calculation formula

is

$$SNR = \frac{\overline{U}_f}{\overline{C}} \tag{8}$$

$$\overline{U}_{fa} = \overline{U}_a - \overline{U}_{dark} \tag{9}$$

$\overline{U}_{fa}$ of a single pixel is obtained from Formula 9, $\overline{U}_a$ is the average value of multiple measurements

of the a-th pixel under lighting conditions with a certain integration time, and $\overline{U}_{dark}$ is the average value

of the dark background measured without light under the same condition. The average value of signal

jitter $\overline{C}_a$ is represented by the RMS of the jitter values measured multiple times by a single pixel, which

satisfies

$$\overline{C}_a = \sqrt{\frac{1}{N}\sum_N (U_{ai} - \overline{U}_a)^2} \tag{10}$$

N is the number of measurements used to calculate the average, and $U_{ai}$ is the i-th measurement value

of the a- th pixel. Substituting equations 5 and 6 into 7, the signal-to-noise ratio of a single pixel $SNR_a$

is

$$SNR_a = \frac{\overline{U}_a - \overline{U}_{dark}}{\sqrt{\frac{1}{N}\sum_N (U_{ai} - \overline{U}_a)^2}} \tag{11}$$

From the above analysis, it is known that the SNR of the fiber spectrometer is related to the degree of

pixel exposure. Therefore, the SNR calculation is based on the degree of exposure achieved during the

actual data collection process. The pixels were analyzed according to the level of exposure attained

during the actual collection process. A broadband light source was used to scan multiple times in both



clear and dark conditions to obtain the pixel data for exposed and non-exposed areas, respectively.

Formula 8 is used to calculate the SNR of the pixels in the illuminated area of each binning area. The SNR of the 2-3 lines at the upper and lower edges of 15 lines in each area is lower, ranging between 64.1 and 82.2, while the more concentrated part of the spectrum signal in the middle lines mostly falls between 97.5-123.1. In practical measurement, it should be noted that reducing the number of rows necessitates increasing the integration time to achieve a certain light intensity, which will compromise the time

resolution. Taking comprehensive considerations into account, the binning area in the experiment mostly consists of 10-12 rows. The SNR of each row was higher than 100 to ensure the quality of the collected spectra.

### 4.2.2 Optimized collection of scattered light at pitch angles

The conventional MAX-DOAS system typically employs a motor to rotate the lens to adjust the angle of

spectrum collection. Since only one angle of light enters the spectrometer at a time, each angle does not interfere with the other. It is easier to control the integration time to achieve the same spectral intensity level for all angles(Pinardi et al., 2013). The angles of the lens in this experiment were based on the MAX-DOAS setting(Zhang et al., 2023). The multi- channel combined optical fiber was used to collect spectra of all pitch angles (1°, 2°, 3°, 5°, 8°, 10°, 15°, 30°) simultaneously, However, there

was a significant disparity in light intensity at each angle, making it impossible to ensure uniform light intensity levels at each angle with the same integration time. In DOAS inversion, accurate results are only achieved when the spectrum reaches sufficient light intensity and in order to minimize the cycle time of low angles, it is not suitable to divide low angles into multiple groups. A high-speed electronic shutter controller was also designed in the experiment. The control program divides the pitch angles into

two parts (1°, 2°, 3°, and 5° form one group, and 8°, 10°, 15°, and 30° form another), ensuring that the light intensity of the collected spectrum meets the inversion requirements.

### 5.Resluts and discussion

### 5.1Spectal retrieval

The FS MAX-DOAS system achieves continuous operation through the automatic acquisition program.

The basic strategy is to first control the integration time to achieve the desired light intensity conditions and then adjust the averaging time to ensure a constant total acquisition time for each spectrum. The integration time is affected by weather and lighting conditions. Spectra with a long integration time and poor quality must be excluded during the actual spectrum processing. All measured spectra were



subjected to bias and dark current correction(Chan et al., 2019) and analyzed using the QDOAS software

(http://uv-vis.aeronomie.be/software/QDOAS/). The fitting wavebands of $NO_2$ and HCHO were 338-370

and 336.5-359nm respectively. This study used the zenith spectrum ($\alpha = 90°$) at noon each day as the

reference spectrum for analyzing dSCDs. A low-order polynomial was added to the fitting to eliminate

broadband structures caused by Rayleigh and Mie scattering. The setting for the DOAS inversion of these

gases are listed in Table 4. Fig 7 shows an example of the DOAS fitting of $NO_2$ and HCHO in the

spectrum at a $15°$ observation angle on October 24, 2023, The dSCD of $NO_2$ is $6.78 \times 10^{16}$ molec/cm$^2$,

RMS = $6.29 \times 10^{-4}$, while the dSCD of HCHO is $1.89 \times 10^{16}$ molec/cm$^2$, RMS=$6.11 \times 10^{-4}$.

**Table 4 Parameter configurations of QDOAS inversion $NO_2$, HCHO, " √ "means the parameters used in the inversion**.

| Parameter | Data Source | Species | |
|---|---|---|---|
| | | **NO₂** | **HCHO** |
| Wavelength range(nm) | | 338-370 | 336.5-359 |
| NO₂ | 220 and 298K, Io-corrected, (Vandaele et al,1998) | √ | √ |
| O₃ | 223 and 293K, Io-corrected, (Serdyuchenko et al,2014) | √ | √ |
| O₄ | 293K, (Thalman and Volkamer, 2013) | √ | √ |
| HCHO | 297K,(Meller and Moortgat,2000) | √ | √ |
| BrO | 223K,(Fleischmann et al,2004) | √ | √ |
| Ring | | Calculated by QDOAS | |
| Polynomial degree | | 5 | 5 |
| Intensity offset | | Polynomial of order 1(two coefficients) | |



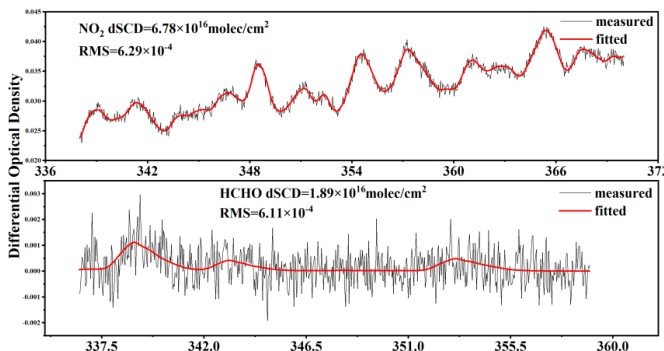

**Figure 7 NO₂, HCHO fitting example(15°)**

## 5.2 Comparison of NO₂ and HCHO MAX-DOAS dSCDs

The FS MAX-DOAS is located on the seventh floor of the Laboratory Building at the Hefei Institutes of Physical Science, Chinese Academy of Sciences. A comparative test was carried out on May 16-18, 2024, using the motor-scanned ground-based MAX-DOAS. Both sets of equipment were oriented to the south, and NO₂ and HCHO were measured simultaneously. The specific positions are shown in Fig 8. The weather was good and cloudless during the experiment, typical of early summer, and the lighting conditions varied greatly throughout the day. Before the actual inversion, the low-quality spectra with excessively long integration time were first removed. Subsequently, the QDOAS results with high RMS values and abnormal concentration fluctuation values were deleted. The data was mostly concentrated between 9:00 AM and 4:00 PM during the day.

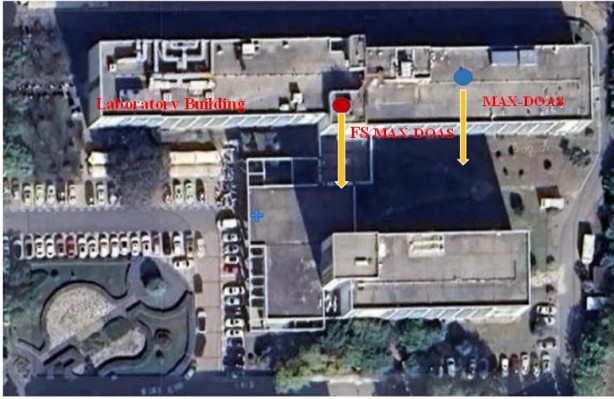

**Figure 8 The specific location of the FS MAX-DOAS and MAX-DOAS (from © Google Maps)**

Figure 9 and 10 show the RMS, dSCD errors, dSCD trends and correlation of NO₂ and HCHO retrieved from spectra collected at several angles by FS MAX-DOAS and MAX-DOAS during the observation



period. The FS MAX-DOAS collected the data at each angle simultaneously, Although two-step

processing was performed at low angles, one cycle was essentially completed in approximately one and

a half minutes. In contrast, the MAX-DOAS system uses a motor to scan a single scope, with one cycle

lasting for eight minutes (the exact time may vary slightly depending on the lighting conditions). For the

sake of comparison convenience, the two sets of concentration data were averaged over a 20-minute

period. Generally, the correlation coefficient of $NO_2$ is higher than that of HCHO, and R exceeds 0.9.

The correlation coefficient of the two sets of data at a 3° elevation angle on May 17th reached 0.978.

The concentration differences of $NO_2$ between the two devices were also smaller. The data correlation

coefficient of HCHO was mostly between 0.76 and 0.85, but the correlation coefficient of the

concentration data at 8° on May 17th reached 0.887. The RMS and dSCD errors can be used as indices

to measure the accuracy of the inversion of the DOAS dSCD. It can also be seen from the boxplot

comparison of the two results that the overall RMS and dSCD errors of the FS MAX-DOAS inversion

of $NO_2$ and HCHO are lower than those of MAX-DOAS. It also demonstrates that the inversion results

of FS MAX-DOAS are more stable and accurate. In general, the HCHO dSCDs detected by the FS MAX-

DOAS were lower than those detected by the MAX-DOAS. From the perspective of the concentration

change trend, both $NO_2$ and HCHO showed that the dSCD values inverted by the FS MAX-DOAS

changed more smoothly compared to the MAX-DOAS data, which exhibited more fluctuations. The

difference can be attributed to the lower temporal resolution of MAX-DOAS and the increased data

fluctuations after time-average processing.

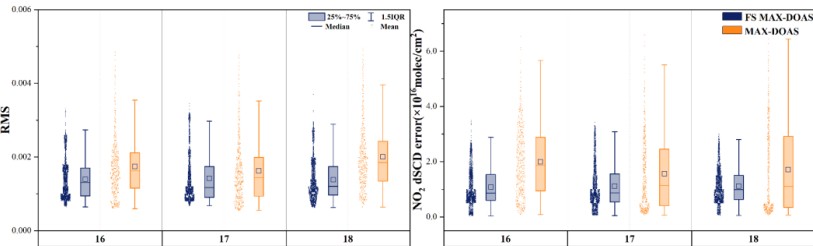



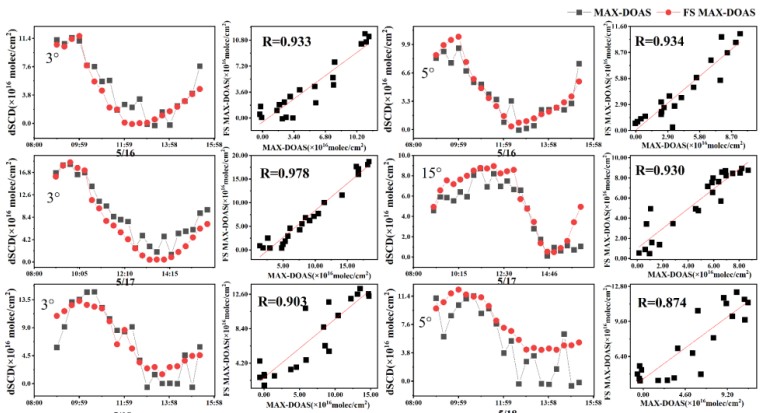

**Figure 9 Comparison of NO₂ dSCDs between FS MAX-DOAS and MAX-DOAS (20 minutes**

**average processing)**

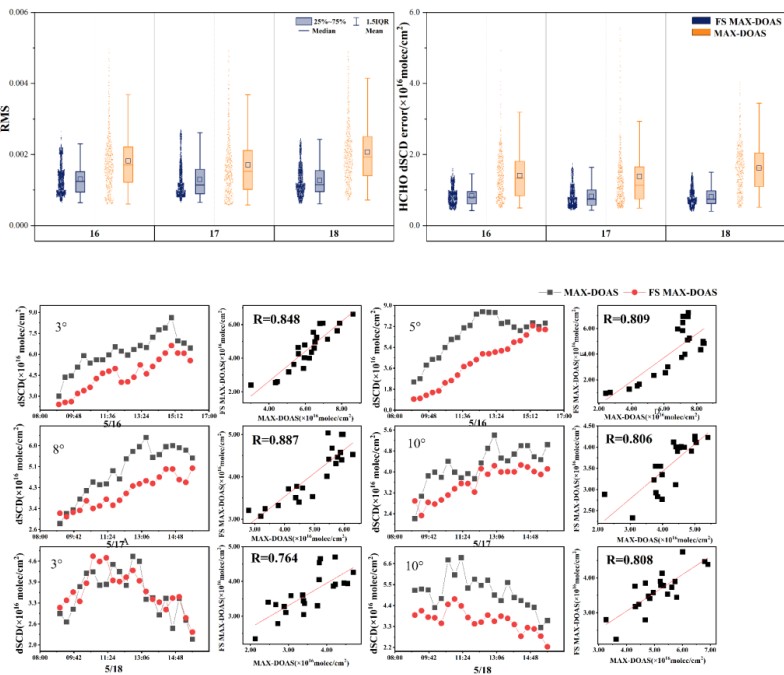

**Figure 10 Comparison of HCHO dSCDs between FS MAX-DOAS and MAX-DOAS (20 minutes**

**average processing)**

**5.3 Vertical profiles of NO₂ and HCHO**

The PriAM algorithm was used to further analyze the trace gas slant column concentration obtained in

this observational experiment. Four kilometers of vertical profiles of NO₂ and HCHO were generated, as



shown in Fig 11. NO$_2$ exhibited significant pollution levels throughout the day from the 17th to the 18th

during the observation period and was transported vertically. The concentration levels was lower than

those in autumn and winter. However, the high concentration of HCHO in early summer was related to

the photochemical oxidation promoted by strong solar radiation and high temperature. From the 16th to

the 18th, there were consecutive sunny days. A continuous process of pollution accumulation can be

observed in Fig 11, where the vertical distribution of HCHO was generally lower than that of NO$_2$. Figure

12 shows a comparison of the original vertical profiles of HCHO obtained by the two systems on May

17th. The trend of change was consistent during the time period from 9:00 am to 12:00 pm. Since FS

MAX-DOAS significantly improves the time resolution, the profile on the right is more detailed than the

one on the left. However, the concentration value of the profile measured by the new method was slightly

395      different from that of MAX-DOAS within a height of 1km, typically varying by around 8%. Figure 13

shows the comparison of a single vertical distribution of NO$_2$ measured by the two systems at 10:23 on

the 17th, Both distributions follow an exponential pattern, and the correlation coefficient between them

is 0.987.

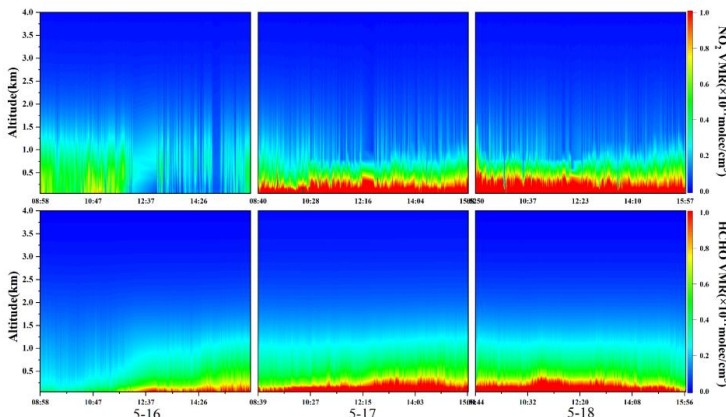

400     **Figure 11 NO$_2$, HCHO profiles obtained by FS MAX-DOAS**

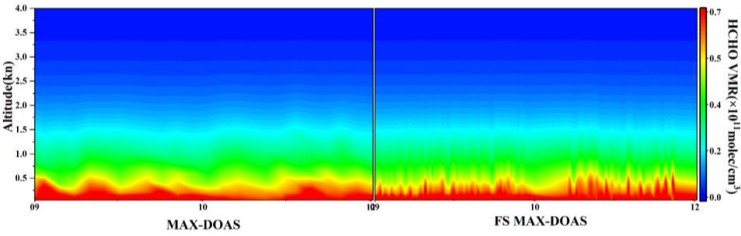





**Figure 12 Comparison of FS MAX-DOAS and MAX-DOAS profiles on May 17 (HCHO, origin)**

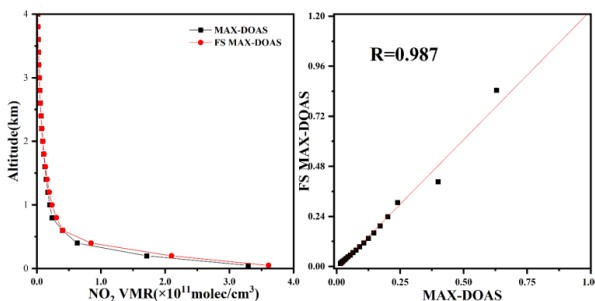

**Figure 13 Comparison of a single profile of NO₂ at 10:23 on May 17**

**5.4 NO₂ surface Volume Mixing Ratios (VMR) comparison of LP-DOAS**

During the measurement period of this experiment, an LP-DOAS was placed on the sixth floor of the Laboratory Building for long-term operation, It can measure $NO_2$, $SO_2$, HCHO, and other gases. Owing to the abnormal data of HCHO measured by LP-DOAS, Fig 15 only compares the near-surface VMR of $NO_2$ from FS MAX-DOAS and LP-DOAS. Generally, the concentration change trends of the two instruments were consistent, showing obvious diurnal variation characteristics of being high in the morning and starting to decrease at noon, The near-surface VMR of LP-DOAS was generally higher than that of FS MAX-DOAS. This LP -DOAS measured the average value within the 700-meter optical path between the telescope and the reflector, specific instrument location is shown in Fig 14. MAX-DOAS usually measures the average value over an effective optical path of approximately 10 km, and variations in the measured air mass lead to specific systematic discrepancies. The telescope of LP-DOAS was on the sixth floor of the Laboratory Building, the reflector was on the sixth floor of Building One, and there is an Innovation Avenue in the middle. In addition, the high volume of vehicles on this road during morning working hours caused the measured value of LP-DOAS to be much higher than that of FS MAX-DOAS before 10 am. The linear correlations between the $NO_2$ VMR measured by FS MAX-DOAS and MAX-DOAS with LP-DOAS were compared. The Pearson correlation coefficients of the two variables were 0.880 and 0.747, respectively, as shown in Fig 16.





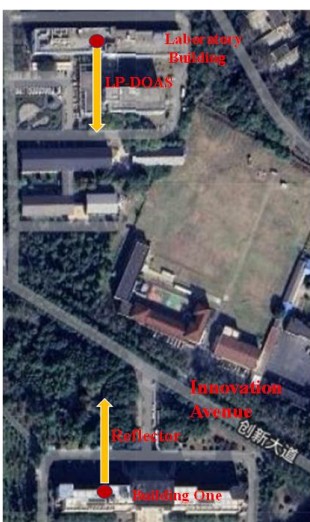

**Figure 14 LP-DOAS telescope, reflector position (optical path of 700m, from © Google Maps)**

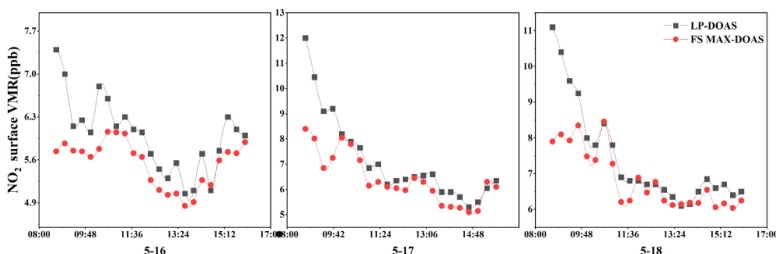

**Figure 15 FS MAX-DOAS NO$_2$ surface VMR comparison with LP-DOAS**

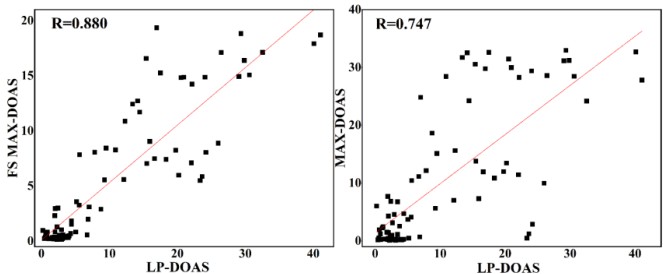

**Figure 16 Correlation of NO$_2$ VMR measured by FS MAX-DOAS, MAX-DOAS with LP-DOAS**

**6. Summary**

This article introduces an experimental system for rapidly acquiring trace gas profiles using multi-

channel spectroscopy. The system controlled the angle of the telescopes as needed through the shutter

switching module to capture scattered light. This light then entered the spectrometer at various locations

on the area array CCD. Subsequently, the light was binned into spectral information with different pitch



angles, significantly enhancing the time resolution of spectral collection. The optimal resolution range

(0.3-0.6 nm) for gas inversion was determined through simulation and analysis of the impact of spectral

resolution on the detection of $NO_2$ and HCHO by FS MAX-DOAS. The system was placed on the seventh

floor of the Laboratory Building to measure $NO_2$ and HCHO in the actual atmosphere and compare the

data with MAX-DOAS and LP-DOAS during the same observation period (May 16-18, 2024). Among

the comparison results of dSCDs at various angles between FS MAX-DOAS and MAX-DOAS, the linear

correlation (R) of $NO_2$ reached 0.9. Specifically, the angles of 3°and 5° exhibited the highest correlation

coefficients. The R for HCHO ranged mostly between 0.76 and 0.85, with 10° and 3° showing the highest

correlation coefficients. The results of the QDOAS inversion showed that the RMS and dSCD errors of

the FS MAX-DOAS spectra inversion consistently stayed lower than those of MAX -DOAS for an

extended period. Owing to the improved temporal resolution, the gas profile obtained by the FS MAX-

DOAS can show more details, and the correlation coefficient of a single $NO_2$ profile at 10:23 on May 17

was 0.987. Compared with the $NO_2$ near-surface concentration measured by the LP-DOAS, both exhibit

a daily variation trend characterized by higher levels in the morning that start to decrease at noon. Due

to the concentrated emissions from vehicles on the road in the morning and the difference in the optical

path of the two systems, the concentration value of LP-DOAS before 10 am was significantly higher. In

terms of correlation coefficients, the Pearson coefficient of FS MAX-DOAS (R=0.880) with LP-DOAS

was higher than that of MAX-DOAS (R=0.747). This study modifies the previous mode of motor

switching scopes and overcomes the limitation of low measurement time resolution. From the perspective

of the dSCD inversion, the gas concentration in the new system was more stable, and the RMS value was

lower. During the experiment, only a few data sources could be compared for data analysis. In future

studies, a variety of data sources will be considered for comparison. This work can also be integrated

with mobile platforms for navigational observation research, which is crucial for achieving mobile MAX-

DOAS profile measurements. It can also be considered for implementing other gas profile detections,

such as $SO_2$, etc.

**Author contribution**

Jiangman Xu: Conceptualization, Methodology, Software, Writing - original draft. Ang Li: Supervision,
Conceptualization, Methodology, Writing - review & editing. Min Qin: Data curation, Resources.
Zhaokun Hu: Methodology, Writing – review & editing. Hairong Zhang: Resources.

**Data availability**

Data underlying the results presented in this paper are not publicly available at this time but may be
obtained from the authors upon reasonable request.



**Acknowledgments**

This work was supported by the National Key Research and Development Project of China
(No:2023YFC3705601) and National Natural Science Foundation of China (No:42105133).

We would like to thank KeTengEdit ([www.ketengedit.com](www.ketengedit.com)) for its linguistic assistance during the
preparation of this manuscript.

**Competing Interest**

The authors declare that they have no conflict of interest.

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
