# Peer review of "Study of NO2 and HCHO vertical profile measurement based"

_EGUsphere, 2024_

## Author Response (AR2)

Dear Editors and Reviewers:

Thank you for your letter and for the reviewers' comments concerning our manuscript entitled "Study of $NO_2$ and HCHO vertical profile measurement based on Fast Synchronous MAX-DOAS" (ID:egusphere-2024-1695). Those comments are all valuable and very helpful for revising and improving our manuscript, as well as the important guiding significance to our researches. We have studied comments carefully and have made correction which we hope meet with approval. The reviewer comments are laid out below in italicized font and specific concerns have been numbered. Our response is given in normal font and changes/additions to the manuscript are explained.

Responds to the reviewers' comments:

**Reviewer #1**

*This manuscript put forward a new way to implement the MAX-DOAS observation model, comprehensively improved the observation system, and the new method greatly improved the time resolution of the acquisition system, which is of great significance to promote the progress of atmospheric measurement technology. The structure of the manuscript is clear, the resolution parameter analysis is provided as the basis of hardware implementation, and the hardware design and processing skills of spectrum acquisition are introduced in detail. The comparison results of field measurement also prove that the new system is of great value for realizing fast profile measurement. I suggest this work can be published in this journal provided that the following concerns are addressed.*

*1.Comment: Why is the optimal spectral resolution range mentioned in the abstract inconsistent with the results analyzed in Section 3.3?*

**1.Reply**: We sincerely thank the reviewer for careful reading and are really sorry for our careless mistakes. In our resubmitted manuscript, we have corrected the optimal spectral resolution "0.2-0.6" into "0.3-0.6" in Section 3.3. Thank you for your next review.

*2.Comment: According to the fiber structure diagram and Table 3, is there two cores in each low angle designed to enhance the light intensity signal? Are the specific data processed in superposition or average?*

**2.Reply:** Thank you for your comment. When the two cores inside a single low angle transmit light, they will illuminate two regions of the CCD that do not interfere with each other, and the two groups of spectra formed are both the spectra of this angle. Due

to the different response ability of the CCD pixel to the optical signal, the two groups of spectra must be different. Two groups of spectral data of a single angle formed in different regions are processed separately during data processing. When selecting dSCDs results, the data with smaller RMS values are retained as the QDOAS inversion results at this Angle. Specific spectral data are neither superimposed nor averaged, they are dealt with separately.

**3.Comment**: *How is the binning technique mentioned in section 2.1 implemented? Write your own algorithms to handle image metadata?*

**3.Reply:** Thank you for your comment. Binning is a feature in the acquisition software that the instrument is configured with, and what we need to do is understand how binning works and effectively combine it with the experimental requirements to meet the acquisition needs. The software binned the signal of multiple lines before their transfer through the serial readout and gain register. Binning can reduce the susceptibility of the signal to the noise in the serial readout and gain register and readout circuit. Furthermore, summation of the signal of multiple pixels increases the signal in the serial register, the larger signal is less susceptible to the noise added in the readout circuit and the standard deviation of the total noise[2], thereby increasing the signal to noise ratio further.

**4.Comment**: *The subscripts in some formulas are too small and not clear enough.*

**4.Reply:** Thanks for your careful checks. We are sorry for our carelessness. Based on your comments, we have made the corrections to harmonize the font sizes throughout the manuscript, and these changes may not be visible in a manuscript in revision mode.

**Reviewer #2**

*At present, in the existing domestic studies, MAX-DOAS system mostly adopts the way of rotating the lens of the motor to convert the observation Angle, which is extremely low efficiency. Some studies also reduce the number of low elevation angles to improve the time resolution, but sacrifice the accuracy of the profile results. This MS discusses the development and implementation of the new MAX-DOAS system, especially the innovation in changing the observation method to improve the temporal resolution has important scientific significance, and can provide more accurate data support for regional pollution monitoring. The technical method of the experiment is reasonable and feasible, which demonstrates the potential application of the method to capture*

*short-term pollution events quickly. In general, this MS is well organized and suitable for publication. Some revision should be made before it can be accepted.*

**1.Comment:** *The introduction should focus on the hardware technical background of MAX-DOAS, and other technologies and algorithms should be appropriately omitted.*

**1.Reply**: We sincerely thank the reviewer for careful reading and useful suggestion, and have re-written this part according to the reviewer's suggestion. The hardware technical background of MAX-DOAS has been stressed, and other technologies and algorithms have been appropriately omitted.

**2.Comment:** *In the spectrum acquisition, how is the specific low elevation Angle and zenith Angle switched, please explain in detail.*

**2.Reply:** Thank you for your comment. We will explain in detail how to switch between low angle and 90° zenith angle. As shown in the system diagram in figure 4 in the manuscript, the shutter control module is used to control the lens shutters which were mounted at the end of the elevation lens to achieve the spectrum acquisition of the required elevation angle. The light transmitted by the Y-type fiber will form spectra of 300-380nm band in different regions of the area array CCD after splitted by the grating. As can be seen from Figure 6 of the structure of the fiber, the spectrum of low elevation angle (1) and corresponding 90° zenith angle (A) will illuminate the same region of the CCD at the same time. Therefore, the computer sends instructions to the control module to alternately open the shutter of the low elevation angle and zenith angle lens, each shutter is controlled separately, without spectral crosstalk.

**3.Comment**: *Line 66: Make sure there are spaces between words and symbols "...Leigh(Leigh et al., 2006)from the University..."*

**3.Reply:** We were really sorry for our careless mistakes. Thank you for your reminder, and we've leave space between word and symbol, as you can see in the revised draft submitted.

**4.Comment**: *There are various formulas in the text, please note that when explaining the parameters of the formulas, the units should be added to increase the readability of the article.*

**4.Reply:** Thank you for your insightful comment. After reviewing similar articles published in Atmospheric Measurement Techniques, we noticed that parameters in equations are often presented without specifying units, as the context and variable

definitions provide sufficient clarity. However, to address your concern, we have added units for parameters where necessary to ensure clarity.

**5.Comment**: *Note that the "°" symbols in several places in the text appear to be in Chinese rather than English, so please replace them. For example, Line 314 "(1°, 2°, 3°, 5°, 8°, 10°, 15°, 30°)", Line 320, etc.*

**5.Reply:** Thank you for your careful checks, we are sorry for our carelessness. Based on your comments, we have made the corrections to make the symbol in English and harmonized within the whole manuscript.

**6.Comment**: *The figure names in the text are both 'Figure' and 'Fig', so please be consistent throughout the text, as required by the journal. For example, 'Figure 7' and 'Fig 7'.*

**6.Reply:** Thank you for your careful checks, we are sorry for our carelessness. The journal's description requirements for the graph is "The abbreviation "Fig." should be used when it appears in running text and should be followed by a number unless it comes at the beginning of a sentence, e.g.: "The results are depicted in Fig. 5. Figure 9 reveals that...".". I've modified them as requested.

**7.Comment**: *Figure 9 and Figure 10: Please mark (a) (b) for multi-figure combinations.*

**7.Reply:** Thank you for your careful checks, we feel sorry for our careless, in our resubmitted manuscript, the marks for multi-figure combinations have been revised.

**8.Comment**: *The font in some of the pictures is too small, so please make it as large as possible to ensure it can be read. For example, Figure 9 and Figure 10.*

**8.Reply:** Thank you for your careful checks, we feel sorry for our careless, in our resubmitted manuscript, we have made the font in those pictures as large as possible to ensure it can be read.

**9.Comment**: *Why is there a big difference in concentration in the trend chart and correlation coefficient chart in Section 5.4 compared with long-path near-ground concentration?*

**9.Reply:** Thank you for your careful checks, we feel sorry for our careless. In fact, we did the experiment for several days, and the results of the concentration trend chart showed the high-quality data of three days when the weather was clear. The correlation

coefficient chart was incorrectly selected due to the unmarked date, and we are very sorry for this error. We have replaced the correlation coefficient chart with the data of these three days.

*10.Comment*: *Consider including further system optimization discussion in the outlook.*
**10.Reply:** Thank you for your careful checks, we think this is an excellent suggestion, and have added further optimization of the system in the outlook section.